# Control Method in Coordinated Balance with the Human Body for Lower-Limb Exoskeleton Rehabilitation Robots

**DOI:** 10.3390/biomimetics10050324

**Published:** 2025-05-16

**Authors:** Li Qin, Zhanyi Xing, Jianghao Wang, Guangtong Lu, Houzhao Ji

**Affiliations:** School of Electrical Engineering, Yanshan University, Qinhuangdao 066012, China; yiaa10249@163.com (Z.X.); a3014954129@163.com (J.W.); 18503385602@163.com (G.L.); jilala6909@163.com (H.J.)

**Keywords:** lower-limb exoskeleton rehabilitation robot, balance coordination, reinforcement learning, supervised learning, optimization

## Abstract

Ground walking training using a floating-base lower-limb exoskeleton rehabilitation robot improves patients’ dynamic balance function, thereby increasing their motor and daily life activity capabilities. We propose a balance-directed motion generator (BDMG) based on the principles of deep reinforcement learning. The reward function sub-components pertaining to physiological guidance and compliant assistance were designed to explore motion instructions that are harmoniously aligned with the human body’s balance correction mechanisms. To address the sparse rewards resulting from the above design, we introduce a stepwise training method that adjusts the reward function to control the model’s training direction and exploration difficulty. Based on the aforementioned generator, we construct a training and evaluation process database and design an abnormal command recognizer by extracting samples with diverse feature characteristics. Furthermore, we develop a sample generation optimizer to search for the optimal action combination within a closed space defined by abnormal commands and extremum points of physiological trajectories, thereby enabling the design of an abnormal instruction corrector. To validate the proposed approach, we implement a training simulation environment in MuJoCo and conduct experiments on the developed lower-limb exoskeleton system.

## 1. Introduction

Balance dysfunction is a key factor that restricts movement and activities of daily living in patients during the post-stroke recovery phase [1,2]. In various balance and mobility scales [3,4], dynamic balance is a high-scoring evaluation item and also an important goal in later-stage walking training. Currently, existing commercial products, such as ReWalk [5], WalkON Suit [6], and HAL [7], assist patients with basic gait rehabilitation training. However, these devices rely on external rigid supports to maintain balance, leading to low patient initiative. In contrast, utilizing a floating-based lower-limb exoskeleton rehabilitation robot for ground walking training, modeled after the assistance techniques used by physical therapists, should offer a more beneficial training approach [8].

Predefined trajectory tracking control is a commonly used control method, which fits human gait data and generates motion trajectories tailored to the patient by considering trunk movement and center of mass changes [9]. Ma, Y. et al. [10] proposed a gait planning method that considers the process of center of mass (COM) shifting. This method uses a finite state machine (FSM) model to switch gait trajectories based on the wearer’s body parameters and posture. Park, K.W. et al. [11] developed an adaptive gait mode adjustment approach that iteratively determines the optimal trunk tilt angle by analyzing ground contact time and subsequently adjusts the joint reference trajectories. Wu, X. et al. [12] introduced a Gaussian process regression-based methodology to generate appropriate gait trajectories according to the patient’s physiological parameters. While these methods enable realistic leg movements and effectively guide the patient’s gait, they constrain human balance responses, thus making them more appropriate for early-stage rehabilitation training using fixed-base exoskeleton robots.

To provide flexible assistance to patients without enforcing step length and step frequency, control methods have been developed to reduce human-robot interaction forces and align with human intent. Caulcrick, C. et al. [13] proposed a model predictive control (MPC) framework that estimates the human-exoskeleton interaction model. This framework utilizes reference angular velocities and joint torques estimated from electromyography (EMG) signals to track human intent. To adjust the human-robot interaction torque, Küçüktabak, E.B. et al. [14] introduced an exoskeleton closed-loop compensation method. This method calculates the interaction torques during the gait cycle by utilizing the measured torques at the exoskeleton’s hip and knee joints, in conjunction with the whole-body dynamics of the exoskeleton.

To further clarify patients’ motor intentions, some researchers have employed fractal dimension methods to quantify gait complexity for assessing patient status and detecting early abnormalities. Dierick et al. [15] applied nonlinear analysis techniques to gait time series, using complexity measures to enhance gait classification accuracy. Chakraborty et al. [16] attempted to differentiate Parkinson’s patients from healthy individuals using fractal dimension, aiming to explore the relationship between the central nervous system and motor function; however, the results were suboptimal, with limited accuracy. In clinical studies, Dehghan et al. [17] found significantly reduced fractal dimensions in several cortical regions of patients with the postural instability and gait difficulty (PIGD) subtype, indicating the potential value of this method in disease identification. Nonetheless, current research remains in the exploratory stage, requiring further validation and standardization.

Some studies have attempted to introduce impedance control to model and predict human motion trajectories. Zhu, A. et al. [18] proposed an impedance control strategy for lower-limb exoskeleton rehabilitation robots based on adaptive force control. This strategy takes the gait of healthy individuals as a reference and implements position control on the exoskeleton, adjusting the gait trajectory according to the patient’s own leg strength. Huang, R. et al. [19] introduced a novel coupled cooperative primitives (CCP) strategy, where a human-robot interaction model, represented using an impedance model, serves as a modulation term. The impedance coefficient of this modulation term is adjusted online to ensure the stability of the human-exoskeleton system. Zhang, T. et al. [20] proposed a hip joint controller based on the extrapolated center of mass. This controller responds to balance disturbances through a combination of a series elastic actuator (SEA) and active compliance control based on adaptive admittance control. By adjusting the hip joint angle, the controller generates compliant guiding forces, ensuring that the zero-moment point (ZMP) remains within the support domain, thereby maintaining the balance of the exoskeleton. Soliman, A.F. et al. [21] proposed a stack-of-tasks approach, in which a priority-based ordering is applied to the task variables of the CoM, left foot, and right foot. A motion controller incorporating ZMP impedance feedback ensures that the ZMP remains within the support domain, thereby maintaining system balance. These methods exhibit rapid responsiveness to changes in interaction forces but still depend on trajectory error correction. Additionally, achieving full compliance may pose a risk of instability and potential falling [22].

To enhance wearer comfort, Martínez, A. et al. [23] proposed a method that allows the user to control the exoskeleton’s step length and timing during the swing phase, avoiding interference with the body’s balance response. Leestma, J.K. et al. [24] designed a hip exoskeleton with both frontal and sagittal plane drives. The exoskeleton’s balance capability is enhanced during steady-state and perturbation walking by adjusting step length and width. Tian, D. et al. [25] introduced a CoM modification controller based on center of mass dynamics. Designed according to human body structure, this controller eliminates axial errors and enhances the exoskeleton’s center of mass stability, achieving self-balancing during walking.

Beck, O.N. et al. conducted an analysis of physiological data, including soleus muscle length, surface electromyography (sEMG) signals, and joint reaction torques, when the human body is subjected to external disturbances. The experimental findings demonstrated that when the exoskeleton provides assistive torque aligned with human intent, applying assistance within the first 50–70 ms before the body’s initial balance correction response increases human-exoskeleton interaction forces but contributes to reducing CoM disturbance and maintaining balance. Conversely, delaying assistance by 50–70 ms reduces interaction forces yet fails to effectively support balance maintenance [26]. Thus, achieving effective coordination with the body’s balance correction response remains a critical challenge for floating-base lower-limb rehabilitation exoskeletons in assisting overground gait training.

Based on the above analysis, an optimal gait controller is designed using deep reinforcement learning to explore how the exoskeleton robot can provide physiologically plausible guidance and compliant assistance in coordination with the human balance correction response.

The main contributions of this study are as follows:(1)A BDMG based on a confidence domain strategy optimization algorithm is designed, incorporating physiological gait trajectories, system balance states, and compliance states into the reward function. To improve the model’s convergence speed, a staged training method is proposed, controlling the model’s training direction and exploration difficulty by adjusting the reward function.(2)Based on the training and evaluation data of the BDMG model, an abnormal command recognizer is designed and trained to identify abnormal commands after the user–exoskeleton system performs actions. To correct these abnormal commands from the BDMG, an abnormal command corrector is designed. The closest point on the physiological gait trajectory to the current joint position is considered the physiological trajectory’s extreme point. The optimal action combination is explored within the closed space formed by the abnormal commands and the physiological trajectory’s extreme point.(3)Based on the optimal action combination data, a supervised learning algorithm is used to train the abnormal command corrector. An overall control framework, consisting of the abnormal command corrector, abnormal command recognizer, and BDMG, is then designed to form the optimal gait controller, achieving end-to-end balance and compliance control for the lower-limb exoskeleton robot.

The structure of this paper is as follows: Section 2 provides a detailed description of the design process of the BDMG to obtain the optimal motion strategy. Section 3 addresses the issue of policy drift and introduces the abnormal command recognizer to identify erroneous commands. Section 4 further designs the abnormal command corrector to correct the abnormal commands. These three components together form the optimal gait controller. In Section 5, the performance of the optimal gait controller in terms of balance and compliance is verified through simulations. Section 6 presents related experiments, comparing the optimal gait controller’s performance with traditional position control and impedance control, thereby validating its superiority in performance. Finally, Section 7 concludes the paper.

## 2. Design of the BDMG

The BDMG constitutes a vital element of the optimal gait controller, playing a pivotal role in shaping the overall system performance. Its design is critical for ensuring balanced and coordinated movement, providing a foundational basis for the development of an effective gait control strategy. Figure 1 illustrates the overall architecture of the optimal gait controller, highlighting the integrated interactions among its key components.

Physiological trajectory is derived from the joint angle data of 14 full-body joints during normal human walking. The specific design is outlined as follows.

### 2.1. Action Space and State Space Design

The expression of the action space is as follows:(1)$$A_{t}=\left\{q_{i}^{\prime },{\dot{q}}_{i}^{\prime }\right\}$$
where, $q_{i}^{\prime }$ and ${\dot{q}}_{i}^{\prime }$ represent the position and velocity values, respectively, of each joint degree of freedom in the human body at time t, where i∈[1,30]. Among these, the action space formed by the knees and hips of both legs constitutes the physically actuated action space, whereas the remaining elements form the virtual action space.

The expression of the action space is as follows:(2)$$\begin{array}{c} S_{t}=\{S_{c},S_{r}\}\\ S_{c}=\{{\hat{q}}_{i}\}\\ S_{r}=\{q_{i},{\dot{q}}_{i},p_{b},{\dot{p}}_{b},\omega_{b},p_{m},p_{u},b,f_{x},F\} \end{array}$$
where, $S_{c}$ represents the physiological trajectory information, which is the current full-body coordinated optimal value determined based on the action from the physiological trajectory cluster; ${\hat{q}}_{i}$ represents the angle of the i-th degree of freedom at time t; $S_{r}$ represents the actual joint position information of the lower-limb exoskeleton human-machine system, referred to as real-time state information. $q_{i}$ represents the real-time position of the i-th joint of the human body; ${\dot{q}}_{i}$ represents the real-time velocity of the i-th joint of the human body; $p_{b}$, ${\dot{p}}_{b}$ and $\omega_{b}$ represent the position, velocity, and angular velocity at the pelvis of the exoskeleton, respectively; $p_{m}$ represents the ZMP position coordinates of the exoskeleton; $p_{u}$ represents the center coordinates of the support domain at the sole of the exoskeleton; and b is a Boolean value that represents the relationship between the ZMP and the support domain. When the ZMP is within the support domain, b=1; otherwise, b=0. $f_{x}$ represents the scalar value of the human-machine interaction force; F=0 represents the right thigh; F=1 represents the left thigh; F=2 represents the right lower leg; and F=3 represents the left lower leg.

### 2.2. Reward Function Design

The reward function comprises four key components: physiological reward, forward progress reward, balance reward, and compliance reward. These components collectively characterize the maximum reward attainable when the exoskeleton’s physiological guidance, compliance assistance, and human balance correction responses are effectively coordinated.

The physiological reward is designed to enable the human-machine system to converge toward the human physiological trajectory with the greatest possible efficiency. It is defined as follows:(3)$$r_{h}=\exp(-0.5*\sum_{i=1}^{30}\left|q_{i}-{\hat{q}}_{i}\right|)$$

The forward progress reward is designed to incentivize the human-machine system to advance. It is defined as follows:(4)$$r_{f}=0.5*(x_{b.t}-x_{b.t-1})$$
where $x_{b.t}$ represents the x-direction coordinate value of the human pelvis position at time t.

The balance reward aims to improve the balance ability of the human-machine system. It is defined as follows:(5)$$r_{b}=\left\{\begin{array}{cc} 1 & b=1\;\mathrm{and}\;r_{f}>0.005\\ 0 & \mathrm{else} \end{array}\right.$$
where $r_{f}>0.005$, and simulation results confirm that the model exhibits stable and satisfactory performance. This threshold is critical for preventing the lower-limb exoskeleton rehabilitation robot from experiencing balance-related stagnation before acquiring sufficient locomotor capability.

The compliance reward is designed to ensure that the exoskeleton joints move in alignment with the user’s intent, thereby minimizing the interaction force and enhancing the compliance capability of the human-machine system. It is defined as follows:(6)$$r_{s}=\left\{\begin{array}{cc} 1 & f_{x}<f_{\min}\\ {\left((f_{\max}-f_{x})/f_{\max}\right)}^{3} & \mathrm{else} \end{array}\right.$$
where, $f_{\max}$ represents the maximum interaction force that human joints can withstand, with 300 N set for the hip joint and 65 N for the knee joint; $f_{\min}$ is uniformly set to 10 N.

### 2.3. Network Structure and Parameter Design

In the trust region policy optimization (TRPO) algorithm [27], both the value network and the policy network utilize state space values as inputs. The value network generates the corresponding state function values, whereas the policy network produces the probability distribution over actions. The parameter configurations for both the policy network and the value network are presented in Table 1.

The output of the policy network consists of 60 neurons, corresponding to the target positions and velocities of the 30 actuated joints in the human-machine system, while the output of the value network is a single neuron, representing the cumulative expected reward of the current input state.

### 2.4. Stepwise Training Method

Due to the high dimensionality of the state and action spaces in the BDMG, the training process presents considerable challenges and may lead to the agent becoming trapped in a local optimum. Therefore, the stepwise training method is employed to systematically escalate the training difficulty. The training process of the BDMG model is structured into four sequential phases, where the training of each subsequent phase commences upon the convergence of the preceding model. The final reward function is presented in Table 2.

During the training process, the reward function evolves through four stages to gradually enhance the performance of the human-machine system, avoid model overfitting and undergeneralization, and improve training efficiency.
Training 1: The reward function includes two components—forward reward and physiological reward. The goal is to enable the system to achieve physiologically reasonable walking on a smooth surface with a ground friction coefficient of 1. The model converged after 2000 episodes.Training 2: Based on the previous stage, a balance reward is introduced to enhance system stability while maintaining natural gait characteristics. The model converged after 4800 episodes.Training 3: Building on the previous stage, the ground friction coefficient is increased to 1.7 to allow the model to adapt to surfaces that more closely resemble real-world walking conditions, thereby improving robustness. The model converged after 1500 episodes.Training 4: On top of the prior training structure, a compliance reward is added, and the weight of the physiological reward is reduced. This encourages the model to improve compliance during rehabilitation training while maintaining gait balance and physiological plausibility. The final model converged after 4500 episodes.

## 3. Abnormal Command Recognizer

During the simulation and control of the human-machine system utilizing the BDMG, it was observed that the controller could encounter abnormal states, such as transient imbalances or interaction forces exceeding the tolerable limits of the human body. To address this issue, training and evaluation data were extracted from the BDMG, and an abnormal command recognizer was subsequently designed and trained. This recognizer aims to predict the balance and compliance states of the human-machine system following the execution of an action. The detailed design is presented as follows.

### 3.1. Balance Prediction Network Training

Training of the prediction network employs supervised learning algorithms to train both balance prediction and compliance prediction networks. Specifically, the balance sample set and the compliance sample set are utilized as the respective training and testing datasets for these networks. The balance sample set is extracted from the training data of Training 1 through Training 4, as well as the evaluation data of the BDMG. In contrast, the compliance sample set originates from the training data of Training 4 and the evaluation data of the BDMG. Network parameters are updated via the backpropagation algorithm during the training process. The overall training framework is illustrated in Figure 1B.

To ensure the generalizability and robustness of the trained network model, a comprehensive dataset comprising 450,000 data samples was collected across the four training phases. This dataset served as the training input for the balance prediction network, as detailed in Table 3.

The sampling period refers to the interval of time steps at which data are collected. Each data sample includes the balance-related state $M_{B}$, the command values related to the BDMG X, and the balance state at the next time step $g_{t}$. The balance-related state and the command values of the BDMG serve as input features for the balance prediction network, while the balance state at the next time step is used as the label. The balance-related state $M_{B}$ includes the ZMP position, the coordinates of the support domain center, the number of elements in the support domain convex hull, and the positions of all joints in the human-machine system. The command values from the BDMG X represent the motor position command values for the knees and hips of both legs in the sagittal plane.

The hyperparameters utilized for training the balance prediction network are presented in Table 4.

### 3.2. Compliance Prediction Network Training

Training sample dataset for the compliance prediction network is collected from the training 4 process and the model evaluation process. During the training 4 process, the data collection period is 30, with 250,000 samples, while in the model evaluation process, the data collection period is 1, with 100,000 samples. Each data sample includes the compliance-related state $M_{S}$, the command values related to the BDMG X, and the compliance reward value at the next time step $b_{t}$. The compliance-related state $M_{S}$ includes the interaction force vector, the interaction force acting leg identifier, and the coordinates of the middle position between the thigh and calf of both legs in the local coordinate system. The compliance-related state and the command values of the BDMG serve as input features for the compliance prediction network, while the scalar value of the interaction force at the next time step serves as the output label. Compliance prediction network training hyperparameters are shown in Table 5.

During the training process of both the balance prediction network and the compliance prediction network, to ensure the generalization ability of the final model, the sample sets are randomly selected. Additionally, the selected input features undergo normalization to improve the stability and efficiency of the training process, ensuring consistent and reliable model performance across various conditions

## 4. Abnormal Command Corrector

In the optimization of abnormal commands, the PSO algorithm was selected. As a swarm intelligence algorithm, PSO can efficiently explore complex solution spaces without the need for gradient information, making it particularly suitable for the sparse and non-differentiable reward environment in this study. Based on this, a sample generation optimizer is developed using PSO to identify the optimal combination of action samples within the closed space defined by abnormal commands and the extremum points of physiological trajectories. Furthermore, a supervised learning approach is employed to train an abnormal command corrector, which adjusts the transient abnormal commands generated by the BDMG.

### 4.1. Objective Function and Parameter Design of PSO Algorithm

Solution space is defined as Ω and consists of the hip and knee joint position commands of both legs, represented as $X=(x_{1},x_{2},x_{3},x_{4})$, along with the corresponding hip and knee joint position commands and the polar distance point of the physiological trajectory in the BDMG at the same time step, denoted as $Y=(y_{1},y_{2},y_{3},y_{4})$. Where $y_{i}$ represents the polar distance point of the physiological trajectory, which is the point on the physiological trajectory that is closest (in terms of L2 norm) to the current joint positions of the knee and hip joints of both legs in the sagittal plane degrees of freedom of the human-machine system. The mathematical formulation is given as:(7)$$\Omega =\{(Z_{1},Z_{1},Z_{1},Z_{1})|Z_{i}\in [Z_{i\min},Z_{i\max}]\}$$
where $Z_{i\min}=\min(x_{i},y_{i})$, $Z_{i\max}=\max(x_{i},y_{i})$; Z∈Ω denotes the search particles in the PSO algorithm.

In this implementation of the PSO algorithm, each particle updates its velocity and position during the optimization process based on the following formulas:(8)$$\left\{\begin{array}{l} v_{i}^{t+1}=wv_{i}^{t}+c_{1}r_{1}(p_{i}^{t}-x_{i}^{t})+c_{2}r_{2}(g^{t}-x_{i}^{t})\\ x_{i}^{t+1}=x_{i}^{t}+v_{i}^{t+1} \end{array}\right.$$

w is the inertia weight, which controls the influence of a particle’s previous velocity on its current velocity, helping balance global and local search capabilities. $v_{i}^{t}$ represents the velocity of particle i at time t. $c_{1}$ and $c_{2}$ are acceleration coefficients that represent the cognitive (individual) and social (global) learning components, respectively. $r_{1}$ and $r_{2}$ are random numbers uniformly distributed in the range [0, 1], introduced to increase the randomness of the search process. $p_{i}^{t}$ denotes the best position found so far by particle i at time t, $x_{i}^{t}$ is the current position of particle i at time t, and $g^{t}$ is the global best position found by the entire swarm up to time t.

Considering that the objective function in the PSO algorithm should be capable of evaluating the impact of predicted correction commands on the model’s balance and compliance, the prediction network trained in the previous section is adopted as the objective function in PSO to maximize the model’s stability and compliance performance. The objective function $f_{z}$ is defined as follows:(9)$$f_{z}=\omega_{1}R_{b}+\omega_{2}R_{s}$$
where $R_{b}$ and $R_{s}$ denote the objective terms for balance and compliance, respectively. The weighting factors $\omega_{1}$ and $\omega_{2}$ are both set to 0.5. The fitness function $f_{\mathrm{fit}}$ is defined as the negative of the objective function, as follows:(10)$$f_{\mathrm{fit}}=-f_{z}$$

The hyperparameters of the PSO algorithm are set as shown in Table 6.

### 4.2. Abnormal Command Corrector Training

One hundred thousand samples of balance-related state information $M_{B}$, compliance-related state information $M_{S}$, and BDMG commands X were collected at the moment immediately preceding the abnormal command. For each abnormal data sample, the corresponding physiological trajectory polar distance point Y was computed and stored in the abnormal state dataset. The PSO algorithm was then applied to each abnormal data sample in this dataset to optimize the action selection, and the optimized results were stored in the optimal action combination sample set.

After the correction sample dataset has been generated, the abnormal command corrector was trained via a supervised learning algorithm. Specifically, balance-related state information $M_{B}$, compliance-related state information $M_{S}$, BDMG command values X, and physiological trajectory polar distance points Y extracted from the correction sample dataset were utilized as input features for the supervised learning algorithm. Meanwhile, the corrected command values acted as the supervision labels. The hyperparameters employed during the training process are summarized in Table 7.

The loss function is defined as:(11)$$Loss=\frac{1}{128}\times \sum_{i}^{128}\sum_{j=1}^{4}\left|(Z_{\mathrm{ij}}-Z_{\mathrm{ij}}^{\prime })\right|$$
where $Z_{\mathrm{ij}}$ represents the j-th element in the i-th set of corrected position command values within the batch, while $Z_{\mathrm{ij}}^{\prime }$ refers to the corresponding instruction used for training evaluation.

## 5. Simulation Model and Result

A virtual training environment for the human-machine system was developed using the MuJoCo physics engine, and the effectiveness of the optimal gait controller was verified through simulation experiments.

### 5.1. Simulation Model

To accurately obtain training results, a human-machine system model based on MuJoCo was developed. The upper body is modeled to reflect normal human physiological posture and reactions, while the lower body, disregarding the influence of the user’s lower limbs on balance, is represented by a lower-limb exoskeleton rehabilitation robot model as the lower-limb component of the system.

Enhancing the generalizability of the trained policy model requires designing the human-machine system model in the simulation based on the weight distribution of various human body parts. The specific weight distribution settings for each component of the human-machine system model are detailed in Table 8.

Based on the actual distribution of degrees of freedom at each joint in the human body, Table 9 provides a detailed description of the degrees of freedom and joint configurations for each component of the human-machine system model.

### 5.2. Solution of Human-Machine System Interaction Forces

Equipped with a comprehensive array of sensors, the model incorporates position, velocity, and angular velocity sensors at the pelvis; position sensors at the lower-limb hip and knee joints; and tactile, position, and pressure sensors on the foot sole. The pelvic sensors are utilized to estimate the interaction forces between the wearer and the exoskeleton robot, while the foot sole sensors assist in determining the support domain and its center point within the model.

Designing the interaction force model begins with randomly initializing an interaction force at the center of the wearer’s supporting leg. This force is directed toward a desired point where the interaction force is 0 N, which also represents the desired position of the wearer’s supporting leg.

The distance between the hip joint and the center of the thigh to 0.24 m and the distance between the knee joint and the center of the lower leg to 0.19 m were set. Within the local coordinate system of the hip joint, the initial coordinates $\left(x_{u0},y_{u0},z_{u0}\right)$ of the thigh center are (0,0,−0.24). If the real-time coordinates of the thigh center are $\left(x_{u},y_{u},z_{u}\right)$, the hip joint position sensor provides data (α,β,χ), representing the thigh’s clockwise rotation by α degrees around the x-axis, β degrees around the y-axis, and χ degrees around the z-axis. The coordinates of the thigh center can be expressed as:(12)$$\begin{array}{l} \left[\begin{array}{c} x_{u}\\ y_{u}\\ z_{u} \end{array}\right]=\left[\begin{array}{ccc} \cos\chi & -\sin\chi & 0\\ \sin\chi & \cos\chi & 0\\ 0 & 0 & 1 \end{array}\right]\left[\begin{array}{ccc} \cos\beta & 0 & \sin\beta \\ 0 & 1 & 0\\ -\sin\beta & 0 & \cos\beta \end{array}\right]\left[\begin{array}{ccc} 1 & 0 & 0\\ 0 & \cos\alpha & -\sin\alpha \\ 0 & \sin\alpha & \cos\alpha \end{array}\right]\left[\begin{array}{c} x_{u0}\\ y_{u0}\\ z_{u0} \end{array}\right]\\ =\left[\begin{array}{c} -0.24*\sin\beta \\ 0.24*\sin\alpha \cos\beta \\ -0.24*\cos\alpha \cos\beta \end{array}\right] \end{array}$$

Within the local coordinate system of the knee joint, the initial coordinates of the lower leg center are (0,0,−0.19). As the lower leg rotates around the knee joint by angles $\left(\alpha^{\prime },\beta^{\prime },\chi^{\prime }\right)$, the coordinates of the lower leg center $\left(x_{d},y_{d},z_{d}\right)$ can be expressed as:(13)$$\left[\begin{array}{c} x_{d}\\ y_{d}\\ z_{d} \end{array}\right]=\left[\begin{array}{c} -0.19*\sin\beta^{\prime }\\ 0.19*\sin\alpha^{\prime }\cos\beta^{\prime }\\ -0.19*\cos\alpha^{\prime }\cos\beta^{\prime } \end{array}\right]$$

Lower-limb exoskeleton rehabilitation robot platform primarily facilitates rehabilitation training by controlling hip and knee joint movements in the sagittal plane of the wearer, focusing on controlling hip and knee joints along the y-axis. Therefore, when using the lower-limb exoskeleton rehabilitation robot platform, it is only necessary to determine the effective range of interaction force along the x-axis and z-axis for initialization. When the lower-limb exoskeleton’s joints rotate around the y-axis, it forms an arc with the rotating joint as the center and the distance from leg center to joint as the radius. On this arc, four points with the maximum and minimum x-axis and z-axis coordinates are identified. These four points represent expected positions that the center of the user’s leg can reach and are used to initialize the range of interaction force.(14)$$\begin{array}{l} F_{\mathrm{xmax}}=k*(-r*\sin\theta_{\mathrm{xmin}}-x)\\ F_{\mathrm{xmin}}=k*(-r*\sin\theta_{\mathrm{xmax}}-x)\\ F_{\mathrm{zmax}}=k*(-r*\cos\theta_{\mathrm{zmin}}-z)\\ F_{\mathrm{zmin}}=k*(-r*\cos\theta_{\mathrm{zmax}}-z) \end{array}$$
where k represents the strap stiffness coefficient, set to 4000 N/m; r denotes the radius of the arc; x and z are the coordinates of the leg center along the x-axis and z-axis, respectively; $\theta_{\mathrm{xmin}}$ and $\theta_{\mathrm{xmax}}$ correspond to the rotation angles around the axis at the minimum and maximum x-coordinates on the arc, with the clockwise direction defined as positive. Similarly, $\theta_{\mathrm{zmin}}$ and $\theta_{\mathrm{zmax}}$ represent the rotation angles around the axis at the minimum and maximum z-coordinates on the arc. Based on human motion mechanics and reference gait, the thigh-related parameters are set as follows: r=0.24 m, $\theta_{\mathrm{xmax}}=1\;\mathrm{rad}$, $\theta_{\mathrm{xmin}}=-1.57\;\mathrm{rad}$, $\theta_{\mathrm{zmax}}=0\;\mathrm{rad}$, and $\theta_{\mathrm{zmin}}=-1.57\;\mathrm{rad}$. The lower leg-related parameters are set as follows: r=0.19 m, $\theta_{\mathrm{xmax}}=1.57\;\mathrm{rad}$, $\theta_{\mathrm{xmin}}=0\;\mathrm{rad}$, $\theta_{\mathrm{zmax}}=1.57\;\mathrm{rad}$, and $\theta_{\mathrm{zmin}}=0\;\mathrm{rad}$.

Let the interaction force, which is randomly generated in the x-axis direction, be denoted as $F_{r}$. To ensure that the interaction force remains within the valid range, it is defined as follows:(15)$$F_{x}=\left\{\begin{array}{cc} F_{\mathrm{xmin}} & F_{r}\leq F_{\mathrm{xmin}}\\ F_{r} & F_{\mathrm{xmin}}<F_{r}<F_{\mathrm{xmax}}\\ F_{\mathrm{xmax}} & F_{\mathrm{xmax}}\leq F_{r} \end{array}\right.$$

The expected position along the x-axis can be expressed as:(16)$$x_{\mathrm{aim}}=x+F_{x}/4000$$
where x represents the local coordinate value of the center point of the force-bearing leg along the x-axis. Any point (x,z) on the arc satisfies $x^{2}+z^{2}=r^{2}$. Referring to human gait, the range of the hip joint’s rotation angle around the y-axis is [−1.57, 1.57] rad. Therefore, when the coordinate z<0 the target position along the z-axis can be expressed as:(17)$$z_{\mathrm{aim}}=-\sqrt{r^{2}-{x_{\mathrm{aim}}}^{2}}$$

The interaction force along the z-axis can be expressed as:(18)$$F_{z}=4000*(z_{\mathrm{aim}}-z)$$
where z represents the local coordinate value of the force-bearing leg’s center point along the z-axis. Therefore, the initialized interaction force vector is $\left(F_{x},0,F_{z}\right)$.

For the rapid convergence of the trained model, interaction forces are randomly initialized with a probability of 3% during the training process and subsequently updated in accordance with joint movement and position at each time step. The formula is presented as follows:(19)$$\begin{array}{l} F_{x}^{\prime }=4000*(x_{\mathrm{aim}}-x^{\prime })\\ F_{y}^{\prime }=0\\ F_{z}^{\prime }=4000*(z_{\mathrm{aim}}-z^{\prime }) \end{array}$$
where $\left(F_{x}^{\prime },F_{y}^{\prime },F_{z}^{\prime }\right)$ represents the updated interaction force; $x_{\mathrm{aim}}$ and $z_{\mathrm{aim}}$ are the x and z direction coordinate values of the expected position of the wearer’s leg when the interaction force is initialized; $x^{\prime }$ and $z^{\prime }$ are the x and z direction coordinate values of the center of the loaded leg at the current moment, in the coordinate system of its rotating joint.

### 5.3. BDMG Training Results

Training the BDMG consists of four iterations. In each episode, the process evolves from the initial state to the terminal state, accumulating training data in the experience pool. Once the experience pool reaches its maximum capacity, the policy is updated, and the step length, as well as the average reward for each episode, is computed and returned. Specific parameter configurations are outlined in Table 10.

Reward function variation curves over four training sessions are shown in Figure 2. The model converges after 5000 episodes of training.

Conducting a simulation experiment with the trained BDMG serves to validate its physiological plausibility, balance maintenance capabilities, and compliance performance. The joint trajectory curves of the human-machine system model are illustrated in Figure 3.

Figure 3 illustrates the hip and knee joint angle curves across a complete gait cycle. In comparison with the physiological trajectory, the joint curves produced by the BDMG demonstrate a consistent movement trend and gait cycle duration. To further analyze the physiological performance of the BDMG, the Euclidean distance between its joint curves and the corresponding positions of the reference physiological trajectory was computed and visualized in graphical form, as depicted in Figure 4.

Figure 4 indicates that the Euclidean distances of the hip and knee joint points for both legs are primarily concentrated at approximately 0.10 rad and 0.15 rad, respectively. To evaluate the curve fitting accuracy, the mean Euclidean distances between the joint angle curves generated by the BDMG and the physiological trajectory were computed. The results are as follows: right hip joint: 0.0995 rad; left hip joint: 0.106 rad; right knee joint: 0.1552 rad; left knee joint: 0.1506 rad. These results demonstrate that the motion generator exhibits a certain level of physiological guidance capability.

Improvement in the compliance performance of the motion generator is shown in Figure 5.

Directed motion generator models developed from training three and training four controlled the walking of the human-machine system. An identical initial interaction force scalar of 40 N was applied, with the same force-bearing leg. As illustrated in Figure 5, the human-machine interaction forces from training three and training four exhibit similarities; however, the magnitude of the interaction force from training four is smaller compared to that from training three. Notably, at 3 s, the interaction force reduces to 0 N. This indicates that the BDMG trained in training four has achieved a significant enhancement in compliance performance.

### 5.4. Abnormal Command Recognizer Training Results

Training of the prediction network was carried out as described in Section 3. Figure 6a shows the loss function curve and training results of the balance prediction network during the training process. After 80 training iterations, the network completed convergence, with the final single-sample loss function value reaching 0.1. During the same period, the average accuracy of both the training set and the test set increased as the network converged, with the final average accuracy of both the training and test sets maintaining at 95%. Figure 6b shows the loss function curve and training results of the compliance prediction network during the training process. It can be observed that after 60 iterations, the network completed convergence, with the final single-sample loss function value reaching 0.07. During the same period, the average accuracy of both the training set and the test set improved as the network converged, with the final average accuracy of both the training and test sets maintaining at 90%.

For further validation of the abnormal command corrector’s effectiveness, 200 sets of commands generated during the training process in Section 2, along with the corresponding model states at the time, were collected and processed through the abnormal command corrector to improve the abnormal commands. Figure 7 illustrates the changes in balance and compliance metrics before and after correction.

Both metrics are evaluated using the objective functions introduced in Section 4.1 as their respective indicators. The results demonstrate substantial improvements in both balance and compliance states following the correction, particularly in the balance state, which largely satisfies the balance requirements after being corrected and assessed by the prediction network.

## 6. Experiment

The overall structure of the lower-limb exoskeleton hardware platform is illustrated in Figure 8. It primarily consists of a computer terminal, motion control system, motor servo system, sensor data acquisition system, and the exoskeleton itself. The lower-limb exoskeleton robot is equipped with four sets of motor servo systems, which are responsible for actuating the exoskeleton joints. Control strategies are generated by the computer terminal and transmitted to the control system to assist the patient’s movement via joint actuators. During exoskeleton operation, joint states and sensor data are collected in real time and fed back to the terminal for processing.

All participants in this study were healthy adult males, with a height range of (170 ± 2) cm and a weight range of (70 ± 4) kg. The experimental protocol involving human subjects was reviewed and approved by the Ethics Committee of the First Hospital of Qinhuangdao, China.

### 6.1. Abnormal Command Corrector: Effectiveness Analysis

Verification of the corrective effect of the abnormal command corrector on the BDMG was conducted through two experiments. For consistency, the joint position data for the BDMG in both experiments were derived from the optimal gait control strategy and were not processed by the abnormal command corrector. The human-machine interaction force data for the BDMG were computed based on the joint position data and the thigh center coordinates of the user’s load-bearing leg. The balance state of the BDMG was predicted using the balance prediction network.

Experiment 1: The participants wore the lower-limb exoskeleton rehabilitation robot, using the optimal gait control strategy as the control strategy for the robot. The participants walked along a pre-arranged path, and during one gait cycle, when the right leg was lifted approximately 0.15 rad, an interaction force of approximately 60 N was applied downward at the center of the right thigh of the exoskeleton. The angles of four joint motors were collected, real-time human-machine interaction force was calculated, and the system’s balance status was assessed using data from a pressure insole. This experiment aimed to simulate the situation where the BDMG caused an imbalance in the lower-limb exoskeleton rehabilitation robot system, specifically as a result of actions taken to enhance compliance, thereby verifying the correction effect of the Abnormal Command Corrector. The results of the experiment are shown in Figure 9.

As shown in Figure 9, at 0.9 s, the center of the subject’s right hip joint experiences a downward interaction force of 60 N. When using the BDMG, the controller prioritizes overall compliance, which compromises balance and results in a loss of stability. At the same moment, the abnormal command corrector detects and improves the abnormal command. The dashed line from 0.9 s to 1.75 s indicates the active period of the abnormal command corrector, during which it acts to reduce the interaction force while maintaining the subject’s balance. Although the interaction force generated by the corrected command is slightly higher than that from the BDMG alone, it effectively maintains balance during movement. After 1.75 s, no further abnormal commands are generated, and a significant deviation is observed between the motion generator’s command and the actual joint angles, requiring some time for convergence. Overall, the abnormal command corrector demonstrates a clear corrective effect on the output of the BDMG.

Experiment 2: The same experimental procedure as in Experiment 1 was followed. When the subject, wearing the lower-limb exoskeleton rehabilitation robot, lifted the right leg to about 0.45 rad, an approximately 60 N downward human-machine interaction force was applied at the center of the right thigh, and sensor data were collected. This experiment aimed to verify the correction effect of the abnormal command corrector when the BDMG exceeded the human safety threshold of 180 N. The results of the experiment are shown in Figure 10.

Unlike in Experiment 1, at 1.4 s in this gait cycle, the BDMG prioritized overall balance at the expense of compliance, resulting in an interaction force exceeding the predefined threshold of 180 N. At the same moment, the abnormal command corrector received and improved the abnormal command. As shown by the blue dashed line in the human–robot interaction force plot, the corrected command reduced the rate of increase in the interaction force, thereby improving system compliance. After 2.8 s, the interaction force decreased to its minimum level, and no further abnormal commands were generated. Throughout the correction process, the interaction force between the subject and the exoskeleton remained within 205 N, effectively mitigating the abnormal command generated by the BDMG and enhancing compliance without compromising overall balance.

### 6.2. Optimal Gait Control Strategy Performance Verification

Two comparative tests were conducted to verify the performance of the optimal gait control strategy in terms of physiological response, balance maintenance, and compliance.

Test 1: The subject wore the exoskeleton rehabilitation robot and walked while being controlled by three control strategies: traditional position control [28], admittance control [29], and optimal gait control. When the left thigh reached a certain angle, a downward interaction force of approximately 30 N was applied at the center of the left thigh. The purpose of this test was to evaluate how the three control methods handle interaction forces, assessing the performance of the optimal gait control strategy when faced with short-term imbalance or lack of compliance.

Test 2: The subject wore the lower-limb exoskeleton rehabilitation robot and walked for 10 gait cycles at a speed of approximately 0.4 m/s on a pre-arranged track. A human-machine interaction force ranging from 30 to 60 N was randomly applied five times at the centers of the upper and lower leg. The exoskeleton was controlled using traditional position control, admittance control, and optimal gait control strategies. Each control method was tested three times, ensuring that the average initial interaction force applied in each test was approximately 45 N.

In the test, the trajectory for position control was based on the physiological gait trajectory obtained when training the BDMG. Since admittance control requires a high level of autonomous control from the human body to avoid injury to the subject, admittance control was applied only to the joints of the load-bearing leg. Position control was used for the other joints, including the hip and knee joints.

#### 6.2.1. Physiological Performance Evaluation

In Test 1, data for the hip and knee joint angles of both legs, as well as the motor speeds, were collected over one gait cycle. The motor speed variation curves for the hip and knee joints of both legs are shown in Figure 11.

As shown in Figure 11, in the experiment, admittance control was applied only to the joints of the loaded leg, while the hip and knee joints of the other leg were controlled using position control. Therefore, the motor speed variations for the left knee joint, right hip joint, and right knee joint under admittance and position control were identical. Under this motor speed control, the joint angle variations are shown in Figure 12.

As shown in Figure 12, when the left hip joint was raised by approximately 0.32 rad, an interaction force was applied to the center of the left thigh. Upon receiving the interaction force, the optimal gait control strategy reduced the angle of the left hip joint’s upward movement to lower the interaction force. When the left hip joint was raised to 0.45 rad, the hip joint deviated too much from the physiological trajectory. To counteract the imbalance, the optimal gait control strategy increased the hip joint’s upward angle. Under admittance control, with increasing interaction force, the left hip joint’s angle changed significantly, and the interaction force continued to grow as the hip joint rotated downward, eventually reducing the interaction force to 0 N. Position control, on the other hand, did not consider human-machine interaction force throughout the entire gait cycle and forced the human body to walk according to the predetermined trajectory.

For a more accurate evaluation of the physiological performance of the three control methods in response to interaction forces, dynamic time warping (DTW) was applied to compare the left hip joint curve under each control method and the physiological joint trajectory from 0.6 to 3.5 s [30]. A smaller distance between the two curves indicates a higher similarity. Calculation results show that the joint curve under position control had the smallest distance to the physiological joint trajectory, with a value of 0.6148. The optimal gait control strategy followed with a distance of 3.26895. Admittance control had the largest distance of 6.197, indicating the lowest physiological performance.

#### 6.2.2. Balance Performance Evaluation

Foot pressure data were collected during Experiment 1 to assess the balance state of the lower-limb exoskeleton rehabilitation robot system. The specific changes are shown in Figure 13.

In Test 1, at 0.6 s, the interaction force was applied. As shown in Figure 13, with position control, the lower-limb exoskeleton rehabilitation robot system experiences alternating states of balance and imbalance throughout the gait cycle. This is because the target position of position control does not change in response to the interaction force, leading to a conflict between the human joint target position and the exoskeleton joint target position, which causes system imbalance.

Foot pressure data from all time points in Experiment 2 were collected to determine the balance state of the lower-limb exoskeleton rehabilitation robot. The closer the ZMP is to the center of the support polygon, the better the balance performance. For the three different control methods, the proportion of the ZMP falling within 20%, 40%, 60%, 80%, and 100% of the region around the support polygon center is statistically calculated, as shown in Table 11.

From Table 11, it can be seen that the optimal gait control strategy ensures the highest system balance with a maximum of 96.8%, where the proportion of the ZMP in the 20–40% support domain is the largest at 32.8%. Under position control, the maximum proportion of the ZMP in the 40% support domain is 62.2%. Throughout the entire process, admittance control consistently shows the lowest balance gait proportion among the three control methods. Therefore, the optimal gait control strategy has the best balance performance, with 96.7% in the 100% support domain range, followed by position control at 89.2%, while admittance control exhibits the worst balance performance.

#### 6.2.3. Compliance Performance Evaluation

Experiment One involved collecting the motor angle of the exoskeleton’s left hip joint and the position coordinates of the human left thigh center in the local coordinate system. Based on these data, the interaction force was calculated, and the interaction force variation curve is shown in Figure 14.

From Figure 14, it can be observed that at 0.65 s, the interaction force is applied. Since the direction of the interaction force opposes the movement direction of the left hip joint, the interaction force increases under all three control methods. Due to the need to balance compliance and stability, the optimal gait control strategy responds more slowly to the initial interaction force compared to admittance control. However, compared to position control, it significantly reduces the interaction force. Among the three control methods, the optimal gait control strategy has the shortest response time in handling interaction forces, effectively reducing the interaction force to 10 N in a shorter time, demonstrating superior compliance performance.

## 7. Conclusions

Four experimental paradigms inducing human balance correction responses were designed to verify that the optimal gait controller can adaptively maintain physiological guidance and compliant assistance while coordinating with human balance correction responses. The overall training and deployment framework proved effective. The specific conclusions are as follows:The proposed method can guide the human-machine system to achieve a balanced gait while maintaining physiological walking patterns and compliant human-machine interaction. The designed safety mechanism effectively optimizes abnormal commands induced by human balance correction responses, preventing falls and excessive interaction forces beyond the threshold.The virtual training environment built on the MuJoCo physics engine, along with the stepwise training framework, reduces model training time and addresses convergence issues. Additionally, the design of a sample generation optimizer and the method for extracting diverse feature datasets significantly contribute to enriching model prior knowledge and reducing experimental risks.An end-to-end motion instruction generator, combined with a layered deployment architecture, can achieve rapid physical deployment, where the generator does not rely on an ontology dynamics model, and platform compatibility is good.

Future research will further investigate the limitations of using fixed reward weights, which may hinder the adaptability of reinforcement learning strategies to individual differences and dynamic changes in physical function during rehabilitation. To enhance the generalizability and clinical relevance of the proposed method, future studies will incorporate a more diverse participant pool, including female subjects and actual rehabilitation patients, thereby enabling validation in more realistic and complex settings. In addition, we plan to establish separate simulation models for both the user and the exoskeleton robot, allowing for a comprehensive analysis of their interactive dynamics and the optimization of control strategies. Moreover, future efforts will focus on increasing the diversity and generalizability of reference physiological trajectories and developing mechanisms for their real-time adaptive adjustment during training. This will enable the system to better respond to variations in user conditions and support the development of more intelligent and personalized rehabilitation assistance solutions.

## Figures and Tables

**Figure 1 biomimetics-10-00324-f001:**
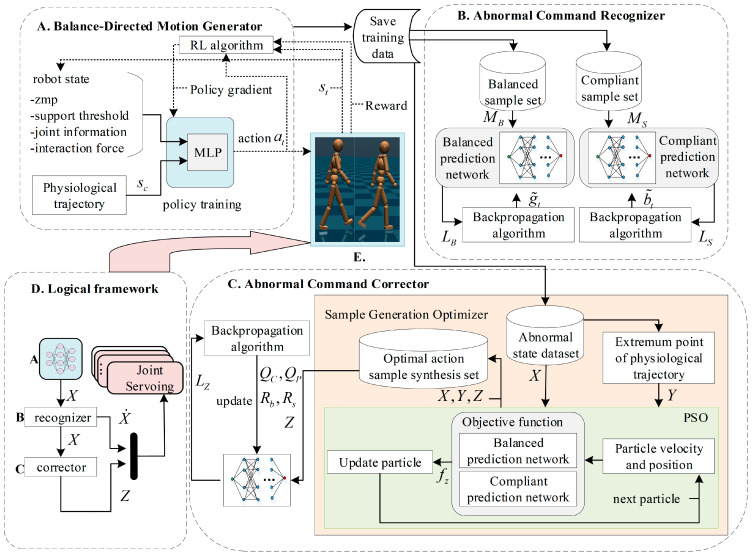
Overall Framework of the Optimal Gait Controller: (**A**) Training the BDMG using RL in simulation. (**B**) Collecting abnormal commands during training to construct the abnormal command recognizer. (**C**) Abnormal command corrector consists of two phases: optimal motion command generation using the particle swarm optimization (PSO) algorithm, followed by training the corrector via data fitting based on existing samples. (**D**) Logical framework flowchart. (**E**) Simulation model.

**Figure 2 biomimetics-10-00324-f002:**
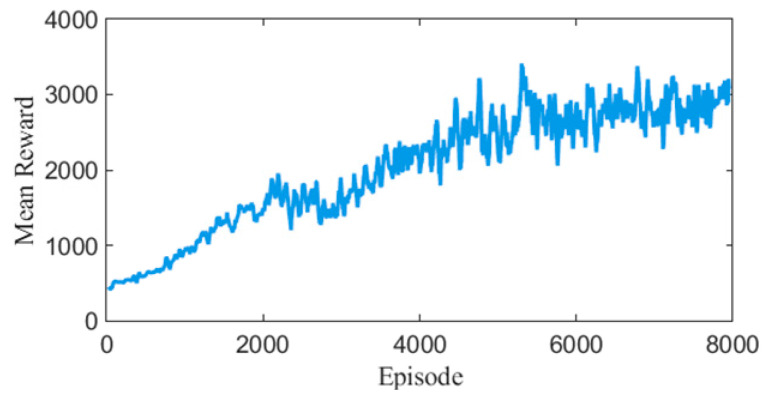
Reward Function Change Curve.

**Figure 3 biomimetics-10-00324-f003:**
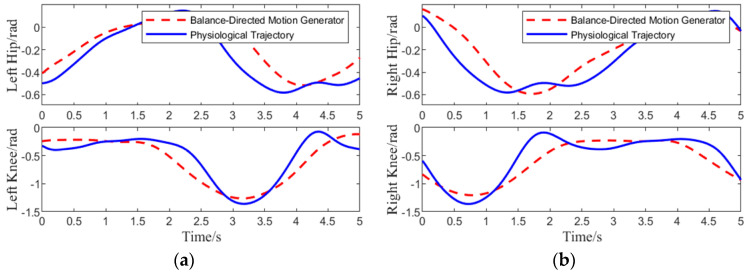
Comparison of Training Results of BDMG: (**a**) joint angle variation curve of the left leg. (**b**) joint angle variation curve of the right.

**Figure 4 biomimetics-10-00324-f004:**
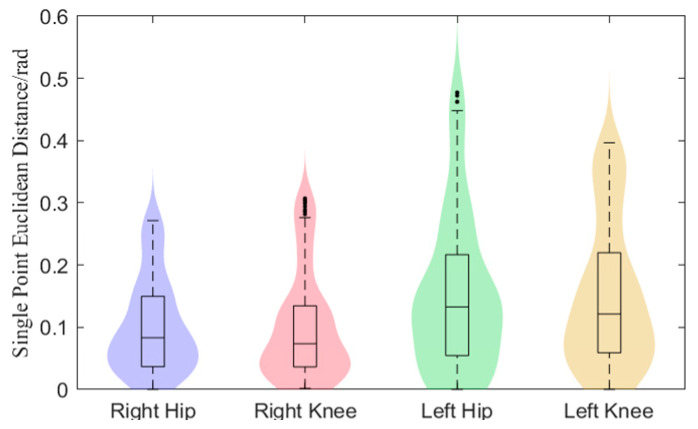
Single Point Euclidean Distance.

**Figure 5 biomimetics-10-00324-f005:**
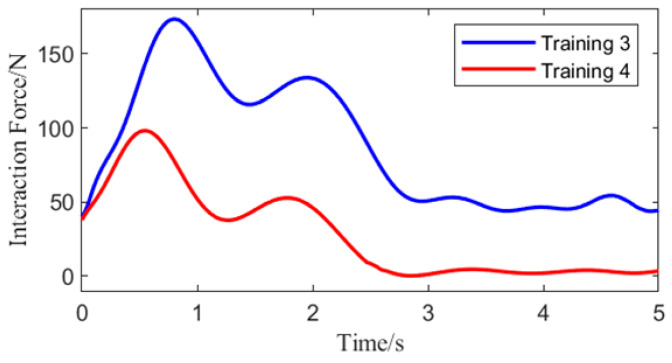
Effect of Compliance Performance Improvement.

**Figure 6 biomimetics-10-00324-f006:**
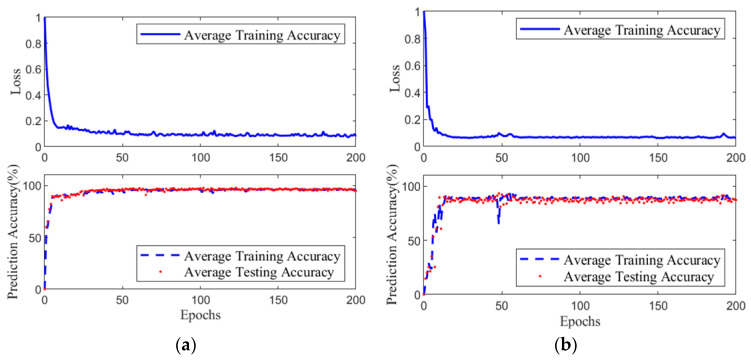
Prediction Results of the Network: (**a**) Variation in the loss function and prediction accuracy for the balance prediction network. (**b**) Variation in the loss function and prediction accuracy for the balance compliance network.

**Figure 7 biomimetics-10-00324-f007:**
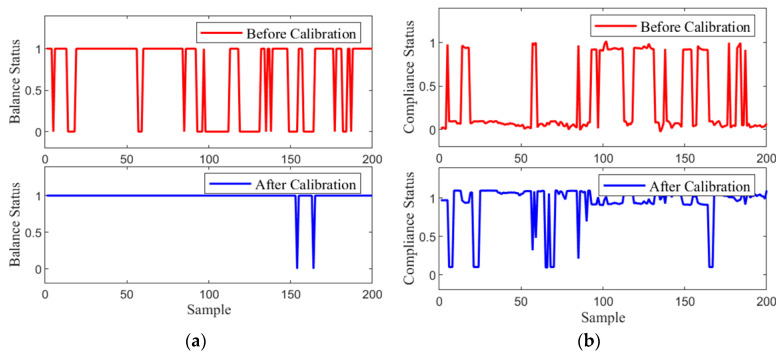
Comparison of the Variation Curves Before and After Abnormal Command Correction: (**a**) The curve of the balance metric before and after correction, where ‘1’ indicates a balanced state and ‘0’ represents an unbalanced state. (**b**) The curve of the compliance metric before and after correction, where ‘1’ indicates a compliant state and ‘0’ represents a non-compliant state.

**Figure 8 biomimetics-10-00324-f008:**
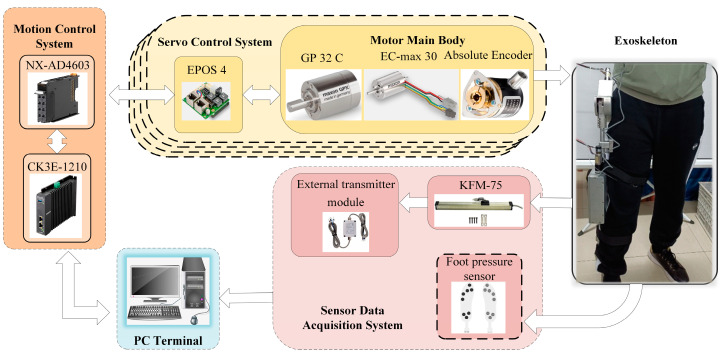
Overall Structure Diagram of the Experimental Platform.

**Figure 9 biomimetics-10-00324-f009:**
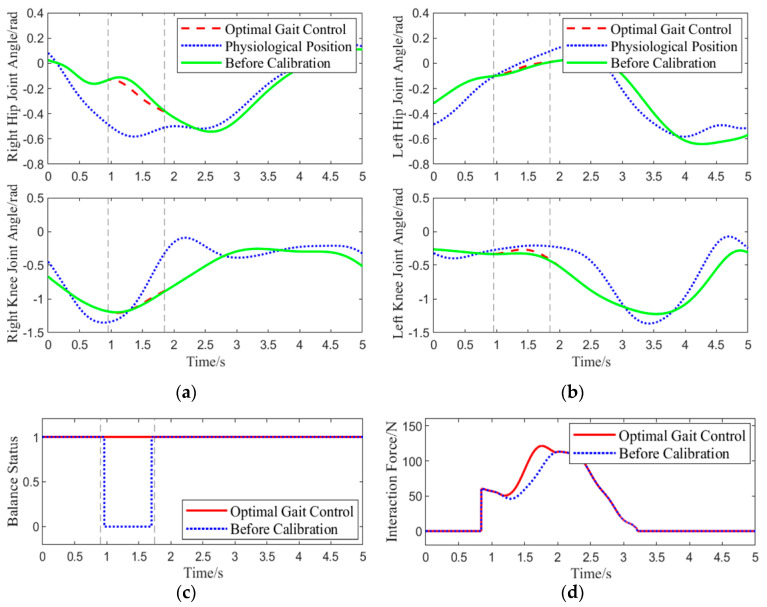
Experiment 1 Gait Motion Optimizer Correction Effect: (**a**,**b**) show the joint angle variation curves for the right and left legs of the exoskeleton, respectively. (**c**) shows the change in the balance state before and after applying the optimal gait control strategy. (**d**) shows the variation in the interaction force scalar before and after applying the optimal gait control strategy. The green line represents the joint trajectories generated by the BDMG; the blue dashed line denotes the reference physiological trajectories; and the red dashed line corresponds to the trajectories corrected by the optimal gait control strategy.

**Figure 10 biomimetics-10-00324-f010:**
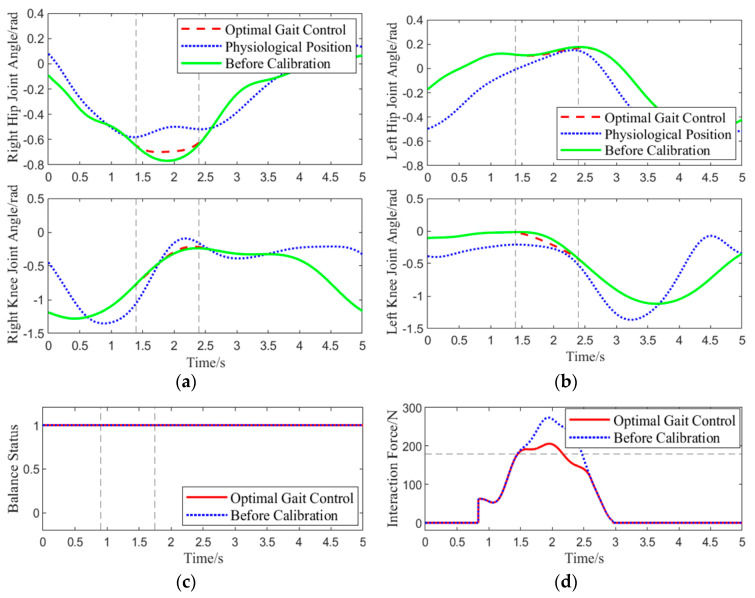
Experiment 2: Gait Motion Optimizer Correction Effect: (**a**,**b**) show the joint angle variation curves for the right and left legs of the exoskeleton, respectively. (**c**) shows the change in the balance state before and after applying the optimal gait control strategy. (**d**) shows the variation in the interaction force scalar before and after applying the optimal gait control strategy. Line representations are consistent with those used in Figure 9.

**Figure 11 biomimetics-10-00324-f011:**
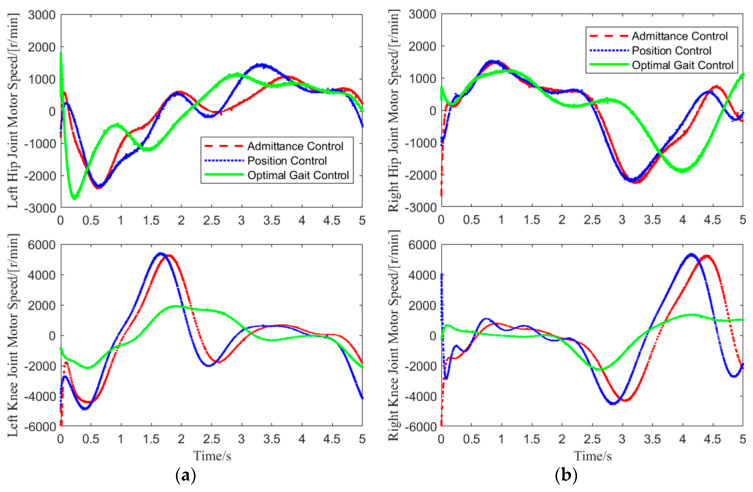
Joint Motor Speed Curve: (**a**) motor speed variation curve of the left leg joint; (**b**) motor speed variation curve of the right leg joint.

**Figure 12 biomimetics-10-00324-f012:**
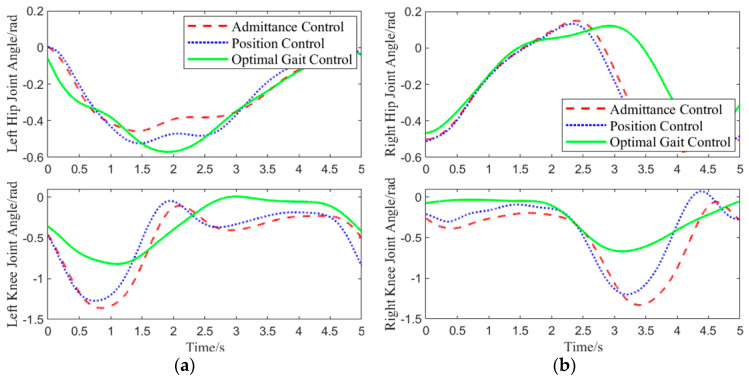
Joint Angle Curve: (**a**) joint angle variation curve of the left leg; (**b**) joint angle variation curve of the right leg.

**Figure 13 biomimetics-10-00324-f013:**
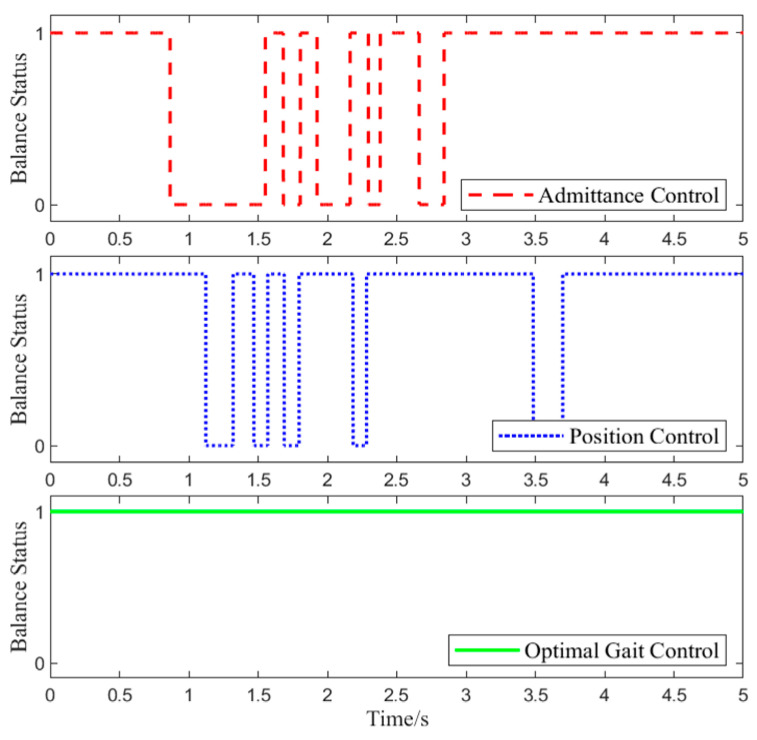
Balance State Variation Curve.

**Figure 14 biomimetics-10-00324-f014:**
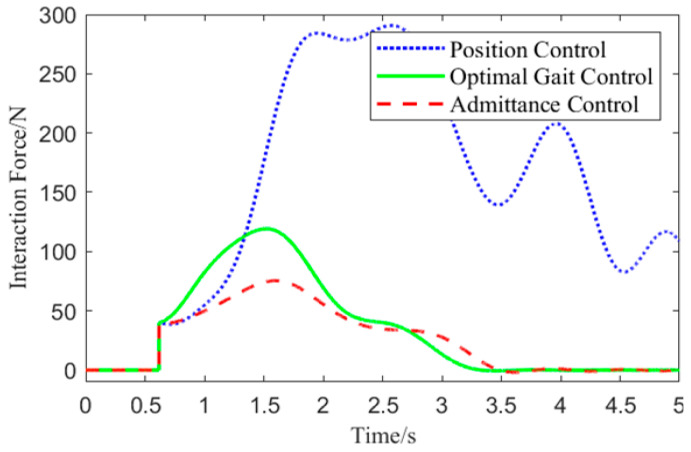
Human-Robot Interaction Force Variation Curve.

**Table 1 biomimetics-10-00324-t001:** Network Parameter Settings.

Strategy Network			Value Network		
Network Layer	Neuron	Activation Function	Network Layer	Neuron	Activation Function
Input Layer	78	-	Input Layer	78	-
Hidden Layer 1	512	tanh	Hidden Layer 1	256	tanh
Hidden Layer 2	512	tanh	Hidden Layer 2	256	tanh
Output Layer	60	-	Output Layer	1	-

**Table 2 biomimetics-10-00324-t002:** Reward Function.

Training Index	Reward Function Formula
Training 1	$R=1.75*r_{h}+0.75*r_{f}$
Training 2	$R=1.75*r_{h}+0.7*r_{f}+0.325*r_{b}$
Training 3	$R=1.75*r_{h}+0.7*r_{f}+0.325*r_{b}$
Training 4	$R=0.75*r_{h}+0.75*r_{f}+0.5*r_{b}+0.5*r_{s}$

**Table 3 biomimetics-10-00324-t003:** Source of Balance Prediction Network Sample Dataset.

Data Source	Sampling Period (Step Size)	Number of Data Samples
Training 1	20	30,000
Training 2	50	70,000
Training 3	50	100,000
Training 4	50	160,000
Evaluation Model	1	90,000

**Table 4 biomimetics-10-00324-t004:** Balance Prediction Network Training Hyperparameters.

Hyperparameters	Parameter Value	Hyperparameters	Parameter Value
Number of Training Samples	360,000	Number of Layers in the Neural Network	90,000
Loss Function	Cross-Entropy	Optimizer	Adam
Learning Rate	0.1	Activation Function	ReLU
Number of Neural Network Layers	4	Number of Neurons	128/256/64/2
Training Epochs	200	Batch Size	256

**Table 5 biomimetics-10-00324-t005:** Compliance Prediction Network Training Hyperparameters.

Hyperparameters	Parameter Value	Hyperparameters	Parameter Value
Number of Training Samples	280,000	Number of Layers in the Neural Network	70,000
Loss Function	Cross-Entropy	Optimizer	Adam
Learning Rate	0.1	Activation Function	ReLU
Number of Neural Network Layers	4	Number of Neurons	256/512/64/1
Training Epochs	200	Batch Size	256

**Table 6 biomimetics-10-00324-t006:** PSO Algorithm Hyperparameters.

Hyperparameters	Parameter Value	Hyperparameters	Parameter Value
Number of Particles	100	Maximum Velocity	0.1
Inertia Weight	0.98	Maximum Iterations	200
Individual Learning Factor	2	Termination Fitness	0.95
Social Learning Factor	2	-	-

**Table 7 biomimetics-10-00324-t007:** Abnormal Command Corrector Training Hyperparameters.

Hyperparameters	Parameter Value	Hyperparameters	Parameter Value
Number of Training Samples	80,000	Number of Layers in the Neural Network	20,000
Loss Function	Cross-Entropy	Optimizer	Adam
Learning Rate	0.1	Activation Function	ReLU
Number of Neural Network Layers	4	Number of Neurons	256/512/128/2
Training Epochs	30	Batch Size	128

**Table 8 biomimetics-10-00324-t008:** Weight Distribution of the Human Body in the Simulation Environment.

Location Names	Total Weight (Human + Exoskeleton) (kg)	Location Names	Total Weight (Human + Exoskeleton) (kg)
Total Weight	(75 + 21)	Forearm	1.0125
Head	3.3	Hand	0.5625
Neck	2.475	Thigh	(7.125 + 1.5)
Shoulder	1.5	Lower Leg	(3.5625 + 1.5)
Chest	10.25	Hip Joint	(0.375 + 2.5)
Abdomen	12.25	Knee Joint	(0.375 + 3)
Pelvis	(10.425 + 1)	Ankle Joint	(0.1875 + 0.5)
Upper Arm	1.8	Foot	(1.275 + 1)

**Table 9 biomimetics-10-00324-t009:** Human Joints and Degrees of Freedom.

Joint Names	Degrees of Freedom	Joint Names	Degrees of Freedom
Root Joint	(x, y, z)	Elbow Joint	(y)
Chest	(x, y, z)	Wrist Joint	(y)
Neck	(x, y, z)	Hip Joint	(x, y, z)
Glenohumeral Joint	(x, y, z)	Knee Joint	(y)
Ankle Joint	(x, y, z)	-	-

**Table 10 biomimetics-10-00324-t010:** Reinforcement Learning Training Hyperparameters.

Hyperparameters	Training 1	Training 2	Training 3	Training 4
Learning Rate 1	0.001	0.001	0.001	0.001
Learning Rate 2	1.5	1.5	1.5	1.5
Discount Factor	0.98	0.98	0.99	0.99
Coefficient of Ground Friction	1	1	1.7	1.7
Experience Buffer Size	6000	6000	6000	6000
Model Update Frequency	20	20	20	20
KL Divergence	0.01	0.01	0.01	0.01
Model Saving Interval	20	20	20	20
Number of Training Steps	665,748	3,556,328	2,024,392	8,844,696

**Table 11 biomimetics-10-00324-t011:** Control Method Balance Performance Comparison.

Support Domain Area	Proportion of ZMP Falling in Different Support Domain Intervals (%)		
	Position Control	Admittance Control	Optimal Gait Control
%20Support Domain	30.2	18.4	25.4
%40Support Domain	62.2	28.1	58.2
%60Support Domain	68.5	53.3	75
%80Support Domain	80.2	73.4	89.9
%100Support Domain	89.2	82.9	96.7

## Data Availability

No new data were created or analyzed in this study.

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
