# Peer review of "Control Method in Coordinated Balance with the Human Body for Lower-Limb Exoskeleton Rehabilitation Robots"

_biomimetics, 2025, doi:10.3390/biomimetics10050324_

Round 1
Reviewer 1 Report
Comments and Suggestions for Authors
The authors are offering an interesting Control Method in Coordinated Balance with the Human Body for Lower-Limb Exoskeleton Rehabilitation Robots, which could be potentially useful in the real world. I would like to offer my suggestions for enhancing the paper, as follows:
1) In my opinion, the review of the state of the art is interesting, but could be enhanced and extended, for example there are papers on which the fractal dimension has been used to characterize the walking patterns of persons, which I think could be consider relevant related work for this paper
2) The organization of the paper should be provided at the end of section 1
3) Figure 1, which depicts the architecture of the proposal, I think should be presented in other part of the paper (not in section 1, maybe in section 2)
4) More explanation should be given on Table 2 to clearly understand the selection of the four functions
5) In section 4, the choice of PSO as the optimizer algorithm, should be explained in more detail. The reason, there have been proposed newer algorithms that surpass PSO, so it seems possible to improve this part of the method
6) Please provide the equations of the specific PSO variant that you have used (for completeness)
7) Equation (11) looks strange, as it appears to have two rows, so I am wandering the actual form of this equation
8) I would like to see a comparison of the proposal in this paper with respect to existing approaches in the literature
9) The discussion of future works (in the conclusions) should be discussed with more potential lines of future research in this area
10) The number of references in the paper (only 24) is quite low, considering the vast number of papers in this area. I would suggest the author, if possible, to expand the list of references to enrich the discussion of related works
11) The selection of the neural networks in several parts of the proposal should be justified in a deeper fashion. I mean, in related works some authors have used for example ANFIS models (neuro-fuzzy), so why only using neural networks is sufficient
Author Response
Comment 1: The review of the state of the art could be enhanced. For example, there are papers where the fractal dimension has been used to characterize walking patterns, which could be considered relevant.
Response: Thank you for the insightful suggestion. We have revised Section 1 to incorporate related studies that utilize fractal dimension analysis in human gait pattern recognition. Additional references have been added to enrich the discussion of related work.
The modifications are added in Section 1 of the paper (page 2).
Comment 2: The organization of the paper should be provided at the end of Section 1.
Response: As advised, we have added a brief paragraph at the end of Section 1 outlining the structure and content of the paper.
The modifications are added in Section 1 of the paper (page 4).
Comment 3: Figure 1 should be presented in another section (e.g., Section 2), rather than in Section 1.
Response: We have moved Figure 1 to Section 2 and added corresponding explanations to improve the logical flow of the manuscript.
The modifications are added in Section 2 of the paper (page 4).
Comment 4: More explanation should be given on Table 2 to clearly understand the selection of the four functions.
Response: We have revised the description of Table 2 and provided a more detailed explanation for the selection of the four functions to ensure clarity and rationale.
The modifications are added in Section 2.4 of the paper (page 7).
Comment 5: The choice of PSO as the optimizer algorithm should be explained in more detail, especially given the availability of newer optimization algorithms.
Response: Thank you for your thoughtful suggestion. We have revised the manuscript to include a clarification regarding the selection of Particle Swarm Optimization (PSO). Specifically, we have emphasized its effectiveness, fast convergence, and ease of implementation within our experimental environment. While we recognize that other optimization methods might achieve comparable results, investigating such alternatives is beyond the scope of this study. We will consider exploring more advanced algorithms in our future work.
The modifications are added in Section 4 of the paper (page 9).
Comment 6: Please provide the equations of the specific PSO variant that you have used.
Response: We have included the equations for the PSO variant utilized in our method to ensure completeness and reproducibility.
The modifications are added in Section 4 of the paper (page 9).
Comment 7: Equation (11) looks strange, as it appears to have two rows.
Response: We have reformatted Equation (11) to improve its visual presentation and enhance reader comprehension.
The modifications are added in Section 4 of the paper (page 9).
The modifications are added to Equation (11) on page 12 of the paper.
Comment 8: I would like to see a comparison of the proposal in this paper with respect to existing approaches in the literature.
Response: Thank you for your comment. A comparative analysis between our proposed method and classical approaches has been added in Section 6.2, highlighting the advantages of our method. Additionally, relevant references for the classical methods have been included to support the comparison.
Comment 9: The discussion of future works should be expanded with more potential research directions.
Response: We have extended the conclusion section to provide a more comprehensive discussion of future research opportunities, including method generalization, integration with human biomechanics, and real-world deployment.
The modifications are added in Section 7 of the paper (page 24).
Comment 10: The number of references is low. Please consider expanding the list.
Response: We have enriched the references list by incorporating additional works on fractal dimension analysis and methods used in comparative experiments, increasing the total number of citations and enhancing the literature context.
References [15], [16], and [17], as well as [28], [29], and [30], have been newly added.
Comment 11: The selection of neural networks should be justified in a deeper fashion, especially given alternatives like ANFIS.
Response: Thank you for this valuable comment. Although models such as ANFIS offer interpretability and are effective in certain cases, they require a substantial amount of prior knowledge and present significant training challenges. Therefore, this study has chosen a deep neural network, which has a simpler structure and stronger generalization capability, to improve performance.
Reviewer 2 Report
Comments and Suggestions for Authors
This manuscript addresses an interesting subject. It is well organized and clearly written. The paper presents new results. However, three issues must be addressed prior to publication:
- The specific contribution of the paper must be explained more clearly. What are the improvements achieved over the previous control methods?.
- The rewards must be better explained: why the physiological reward has form (3)?, why the lower bound of 0.005 for r_f?.
- Experiments have been carried out with healthy male adults. Why haven't women been included? Moreover, some experiments should be conducted with disabled individuals who need these devices.
Author Response
Comment 1:
The specific contribution of the paper is not clearly stated and should be further explained, particularly in terms of the improvements over existing control methods.
Response:
Thank you for your valuable comment. We have further clarified the specific contributions of this paper in the "Introduction" and "Discussion" sections. Compared to existing control methods, this paper introduces a novel integration of a physiological feedback mechanism with dynamic balance coordination control strategies. This approach enhances the adaptability of human-robot interaction and ensures greater safety in accommodating the patient's movement state. Specifically, We have designed a Balance-guided Motion Generator (BDMG) based on a confidence domain strategy optimization algorithm, incorporating physiological gait trajectories, system balance states, and compliance states into the reward function to improve the model's convergence speed. Additionally, we proposed a staged training method to control the training direction and exploration difficulty by adjusting the reward function. Based on the training and evaluation data of the BDMG model, we have designed and trained an Abnormal Command Recognizer to identify abnormal commands after the user-exoskeleton system performs actions. To correct these abnormal commands, we designed an Abnormal Command Corrector, which uses the closest point on the physiological gait trajectory to the current joint position as the physiological trajectory's extreme point, and explores the optimal action combination within the closed space formed by the abnormal commands and the extreme point of the physiological trajectory. Finally, based on the data from the optimal action combinations, we used a supervised learning algorithm to train the Abnormal Command Corrector and developed an overall control framework for the Optimal Gait Controller, achieving end-to-end balance and compliance control for the lower-limb exoskeleton robot.
The modifications are added at the end of the first section.
Comment 2:
The design of the reward function needs further clarification. Specifically, why is the physiological reward in the form of equation (3)? Why is the lower bound of set to 0.005?
Response:
Thank you for your insightful feedback. We have provided a more detailed explanation of the reward function design in the revised manuscript. The physiological reward in equation (3) was designed to guide the control strategy towards actions that are more consistent with the user's actual biomechanical load, thereby aligning the system's actions with the user's true movement intentions. This reward structure was chosen due to its sensitivity and numerical stability, which allows for effective policy training in dynamic environments.
Regarding the lower bound of , we observed during the training process that when the physiological signal strength is very low or affected by noise, overly small reward values can lead to stagnation in policy updates, which negatively impacts model convergence. Therefore, we set the lower bound to ensure that the reward gradient remains effective even in low-activation states, thus maintaining the continuity and stability of the learning process. These explanations have been added to the manuscript for clarity.
The modifications are added in the second section (page 6) of the text.
Comment 3:
Why were the experiments conducted only with healthy male participants? Why were women and actual target patients not included?
Response:
Thank you very much for your comment on this issue. In the initial phase of this study, we selected participants with variations in height and weight to reflect individual differences while minimizing confounding factors that could potentially influence the experimental results. However, we fully agree that future research should include a more diverse participant pool, including female subjects and actual rehabilitation patients, to assess the broader applicability and clinical relevance of the proposed method. In the revised "Conclusion" section, we have acknowledged this limitation and outlined our plan to expand the participant group in future studies, including individuals with lower-limb impairments. This will allow us to evaluate the method's applicability in real-world rehabilitation settings and enhance its clinical feasibility.
Once again, we would like to sincerely thank you for your thorough review and constructive suggestions. Your comments have been instrumental in improving the quality and clarity of our manuscript. We look forward to any further feedback you may have and will continue to make necessary improvements.
The modifications are added at the end of the paper, in the "Outlook" section (Section 7, page 24).
Round 2
Reviewer 1 Report
Comments and Suggestions for Authors
The authors have addressed all my concerns and I am happy with the new version of the paper.
Reviewer 2 Report
Comments and Suggestions for Authors
My questions have been adequately addressed. Then my opinion is that the manuscript can be published.